# The Role of Lymphadenectomy in Early-Stage NSCLC

**DOI:** 10.3390/cancers15143735

**Published:** 2023-07-23

**Authors:** Beatrice Manfredini, Carmelina Cristina Zirafa, Pier Luigi Filosso, Alessandro Stefani, Gaetano Romano, Federico Davini, Franca Melfi

**Affiliations:** 1Division of Thoracic Surgery, Department of Medical and Surgical Sciences, University of Modena and Reggio Emilia, 41121 Modena, Italy; pierluigi.filosso@unimore.it (P.L.F.); alessandro.stefani@unimore.it (A.S.); 2Minimally Invasive and Robotic Thoracic Surgery, Surgical, Medical, Molecular, and Critical Care Pathology Department, University of Pisa, 56126 Pisa, Italy; gaetano.romano@ao-pisa.toscana.it (G.R.); f.davini@ao-pisa.toscana.it (F.D.); franca.melfi@unipi.it (F.M.)

**Keywords:** early-stage, stage I, NSCLC, lymphadenectomy, micrometastases

## Abstract

**Simple Summary:**

The extension of lymph node dissection in the treatment of early-stage non-small cell lung cancer (NSCLC) is currently a controversial argument in the scientific community. There are few studies that look at the role of lymphadenectomy in exclusively stage I NSCLC, and in them the lymph node evaluation method is not standardized, avoiding the comparison between different studies. The future perspectives on this issue will need to focus on the necessity of carrying out more precise investigations into the propagation of micrometastases in lung cancer and the application of the latest available techniques for their detection, resulting in a reduction in both local and distant recurrence.

**Abstract:**

Lung cancer remains the leading cause of cancer-related death worldwide. The involvement of lymph nodes by the tumor has a strong impact on survival of patients. For this reason, lymphadenectomy plays a crucial role in the staging and prognosis of NSCLC, to define the most appropriate therapeutic strategies concerning the stage of the disease. To date, the benefit, in terms of survival, of the different extents of lymphadenectomy remains controversial in the scientific community. It is recognized that metastatic involvement of mediastinal lymph nodes in lung cancer is one of the most significant prognostic factors, in terms of survival, and it is therefore mandatory to identify patients with lymph node metastases who may benefit from adjuvant therapies, to prevent distant disease and local recurrences. The purpose of this review is to evaluate the role of lymphadenectomy in early-stage NSCLC in terms of efficacy and accuracy, comparing systematic, sampling, and lobe-specific lymph node dissection and analyzing the existing critical issue, through a search of the most relevant articles published in the last decades.

## 1. Introduction

Lung cancer is recognized as one of the leading causes of cancer-related death worldwide [1].

Lobectomy with lymph node dissection is the gold standard treatment for patients with resectable NSCLC, as demonstrated by RJ Ginsberg in 1995 [2].

The execution of an adequate lymphadenectomy is essential in lung cancer surgery to obtain an accurate staging and consequently identify patients who could benefit from adjuvant therapies [3]. To date, there are controversies regarding the extent of lymphadenectomy, nomenclature definition, and surgical techniques for intraoperative lymph node evaluation in relation to the stage. In 2006, the Council of the European Association of Thoracic Surgeons (ESTS) published the guidelines for intraoperative lymph node staging in NSCLC [4]. Systematic nodal dissection is recommended in all cases to ensure complete resection, hence all mediastinal tissue containing the lymph nodes is dissected and removed systematically within anatomical landmarks. It is suggested that at least three mediastinal nodal stations (including always subcarinal lymph nodes) should be excised as a minimum requirement. Moreover, the mediastinal nodal stations, the hilar, and the intrapulmonary lymph nodes are dissected as well. Lymph node sampling, though, is defined as the removal of one or more lymph nodes driven by preoperative or intraoperative findings and, according to ESTS guidelines, it should be performed in selected cases in high-risk patients. Systematic mediastinal sampling provides for a predetermined selection of the lymph node stations specified by the surgeons. Lobe-specific systematic nodal dissection is acceptable for peripheral squamous T1 tumors, if hilar and interlobar nodes are negative on frozen section studies. In this type of lymph node dissection, the mediastinal tissue containing specific lymph node stations is excised, depending on the lobar location of the primary tumor (right upper lobe and middle lobe: 2R, 4R, 7; right lower lobe: 4R, 7, 8, 9; left upper lobe: 5, 6, 7; left lower lobe: 7, 8, 9) [4].

In addition, in lung cancer, the minimum number of lymph nodes necessary to consider a lymphadenectomy as complete is not clearly defined. According to NCCN guidelines [5], a minimum of three N2 stations should be sampled or complete lymph node dissection should be performed as a routine in lung cancer resections. Furthermore, formal ipsilateral mediastinal lymph node dissection is indicated in stage IIIA (N2) NSCLC. Instead, ESMO guidelines for the treatment of early and locally advanced NSCLC [6,7] recommend systematic lymph node dissection, conforming to International Association for the Study of Lung Cancer (IASLC) specifications for lung cancer staging [8]. The IASLC has defined systematic nodal dissection as the excision of ≥6 lymph nodes and ≥3 nodal stations, including the subcarinal station [9].

Recently, given the development of the ninth edition TNM staging system, the members of the N-Descriptors Subcommittee of the International Association for the Study of Lung Cancer Staging and Prognostic Factors Committee have reviewed the literature on the different factors to categorize lymph node metastatic involvement, currently based only on anatomical criteria, such as counting lymph nodes, lymph node stations, or lymph node zones, analyzing the pros and cons [10].

To date, the choice of performing lymphadenectomy with an extension influenced by the stage of NSCLC is controversial. There is a critical aspect in clinical practice concerning the assessment of N-status: the necessity of systematic lymph node assessment in early-stage NSCLC.

This review aims to analyze the changes taking place in the execution of lymphadenectomy that could have prognostic implications in the treatment of NSCLC.

## 2. The Role of Lymphadenectomy in Early-Stage NSCLC

In recent decades, given the high incidence of lung cancer, screening programs have been proposed for high-risk patients [11], resulting in the increasing diagnosis of early-stage lung cancer, also thanks to the improvement of staging instrumental investigations [12,13].

As already mentioned, international guidelines define lobectomy with hilum-mediastinal lymphadenectomy as the gold standard treatment for resectable NSCLC. Instead, the extension of lymphadenectomy in early-stage NSCLC is still controversial.

This section summarizes the main results observed in the treatment of early-stage NSCLC with different types of lymphadenectomy, discussing the impact that improved diagnostic-staging pathways are having on the type of treatment needed. We proceeded to a revision of the Medline PUBMED English literature (from January 1994 to May 2023), and we grouped the most relevant studies found according to the design (retrospective, randomized, and prospective non-randomized, and review and meta-analysis), intending to verify the impact of the extension of lymphadenectomy on survival of early-stage NSCLC patients.

### 2.1. Retrospective Studies

In recent years, with the increasing finding of early-stage NSCLC, many surgeons have reported their experience on the proper type of lymph node dissection to achieve the greatest benefits in terms of survival and recurrence. When analyzing the existing literature on studies of a retrospective nature, a targeted search was conducted for articles that only considered early clinical or pathological NSCLC stages, excluding those that also analyzed locally advanced ones i.e., [14], to create a sample as homogenous as possible and to obtain more effective considerations (Table 1).

Since 2003, Wu et al. [15] have proposed the number of totally removed lymph nodes as a prognostic factor in the treatment of pathological stage I NSCLC, validating its use for a more accurate staging with an effect on the survival rate. Moreover, a comparison between complete lymphadenectomy versus sampling in pathological I stage NSCLC showed better overall survival, without enhancing the operative morbidity and mortality, when the more extended lymphadenectomy was performed. In detail, the authors defined complete lymphadenectomy as the removal of at least 10 lymph nodes and a minimum of two mediastinal stations explored [16].

To understand whether complete lymphadenectomy is always necessary for the early stages is essential to identify the possible tumor risk factors associated with lymph node involvement. Veronesi et al. [17] demonstrated that SUV max greater than 2 and pathological tumor size greater than 10 mm are predictive risk factors for lymph node involvement in a population of N0 patients identified by screening. Furthermore, the clinical size of the tumor does not correlate, in contrast to the pathological dimension, with the N status, and therefore the authors suggested the intraoperative measurement of the tumor. According to this study, the systematic lymphadenectomy could be avoided in clinical stage I NSCLC with a preoperative SUV max < 2 at FDG-PET-CT and with a pathological diameter of less than 10 mm.

In recent studies on the role of lymph node dissection in early-stage NSCLC, the use of the preoperative stage was evaluated as a more appropriate factor for clinical practice management. Takizawa et al. published a retrospective study [18] on the effect of lymph node sampling compared to systematic nodal dissection in patients with clinical stage I NSCLC, reporting no significative differences in cancer-specific survival between both techniques.

Stage I NSCLC, according to the eighth edition TNM staging system [19], currently encompasses a broad category of tumors of different sizes, less than 4 cm. On this topic in the literature, there are results on the appropriate lymphadenectomy, stratified by the clinical dimension of the tumor [20], in selected clinical stage IA NSCLC patients: systematic mediastinal lymphadenectomy should be achieved to obtain a potentially better survival in case of lesions between 2 and 3 cm, while lymph node sampling should be performed in patients with lesions of 2 cm or less, with similar results in terms of DFS and OS. More specifically, the size of the tumor was analyzed concerning the number of lymph nodal stations removed in IA NSCLC patients [21]. This study shows that ≥9 overall lymph nodes examined and ≥4 lymph node stations removed are strongly recommended for stage IA2 and IA3, but optional for stage IA1 patients. Therefore, patients in stage IA2 and IA3 NSCLC should undergo systematic lymphadenectomy because it is associated with greater OS and CSS.

More recently, lobe-specific lymphadenectomy has been introduced, motivated by the infrequent detection of mediastinal nodal metastasis in early-stage NSCLC [22] and by the ideally predictable pattern of mediastinal nodal drainage [23,24]. Nevertheless, outcomes related to lobe-specific lymphadenectomy are discordant. To evaluate whether selective lymph node dissection is adequate in the treatment of early-stage NSCLC, Bille et al. analyzed the incidence and distribution of pN1 and pN2 metastases in patients with clinical stage T1-T2N0M0, who underwent systemic mediastinal lymph node dissection [25]. In this study, 16% of patients clinically N0 had pN2 metastases not following a lobe-specific lymphatic drainage pattern. If, therefore, lobe-specific lymphadenectomy had been performed in this series, 16% of patients would have been down-staged, not obtaining a radical surgical treatment and access adjuvant therapies. Moreover, adenocarcinoma histology was found to be an independent risk factor for occult pN2 metastases, as already noted in other published papers [26,27]. According to these results, in conclusion, systematic lymphadenectomy is highly recommended also in the early stages.

**Table 1 cancers-15-03735-t001:** Selected retrospective studies investigating the role of lymphadenectomy in early-stage NSCLC.

Investigators	Year	N of Patients	Stage	Results
Wu et al. [15]	2003	321	pI	The number of removed lymph nodes affects OS and CSS in pathological stage I NSCLC
Doddoli et al. [16]	2005	465	pI	Systematic lymphadenectomy (minimum of 10 lymph nodes assessed and two mediastinal stations sampled) improve OS, not increasing operative mortality in pathological stage I NSCLC
Takizawa et al. [18]	2008	119	cI	Mediastinal lymph node sampling showed a similar diagnostic and therapeutic effect to systematic nodal dissection in terms of CSS in patients with cI NSCLC
Ma et al. [20]	2008	105	cIA	In patients with lesions of 2 cm or less, lymph nodal sampling should be performed with similar effects in terms of DFS and OS, respect to systematic lymphadenectomy
Veronesi et al. [17]	2011	97 + 193	cT1-T1N0M0	Systematic nodal dissection can be avoided in early-stage clinically N0 NSCLC (with max SUV 2.0 or pathological nodule size 10 mm)
Bille et al. [25]	2016	1667	cI	Recommend systematic lymphadenectomy in clinical stage I NSCLC. A total of 16% of patients had upstaging beyond the lobe-specific lymphatic drainage
Zhao et al. [21]	2021	12490	IA	≥9 lymph nodes examined and ≥4 regions of lymph nodes removed are highly recommended for stage IA2 and IA3, but optional for stage IA1

### 2.2. Randomized and Prospective Non-Randomized Studies

In the following years, several prospective non-randomized and randomized clinical trials have evaluated the role of different types of lymphadenectomy in the treatment of early-stage NSCLC (Table 2).

The first controlled prospective clinical trial on the effectiveness of systematic mediastinal lymphadenectomy compared with lymph node sampling in the treatment of resectable NSCLC (cI-IIIA) was published by Izbicki et al. [28] in 1998. In the sampling group, lymph nodes at stations 10, 11, 12, 4, 5, and 7, according to the lymph node mapping of the American Thoracic Society [29], were routinely sampled and the other mediastinal stations from 2 to 9 were also explored, removing any suspicious lymph nodes. In the systematic lymphadenectomy group, a radical systematic en-bloc mediastinal lymphadenectomy, as described by Naruke [30] and Martini [31], was performed. The results of this trial showed that systematic lymphadenectomy does not improve OS and DFS in patients with pathological N0, while it appears to prolong DFS in patients with pathological N1 or N2 of only one mediastinal lymph node station. The authors conclude that in patients with clinical N0 disease, systematic lymphadenectomy does not appear to be necessary, since it does not affect long-distance oncological outcomes. Nevertheless, they still recommend systematic mediastinal lymphadenectomy in patients with resectable NSCLC, given the difficulty with the available diagnostic and staging methods to identify these patients and the non-greater morbidity and mortality [32].

In contrast, a further prospective randomized clinical trial was published by Sugi et al. [33] a few months later, resulting in favor of lymph node sampling. This study included patients with early-stage NSCLC (cT1 <2 cm N0M0) undergoing lobectomy and randomly assigned to lymph node sampling or systematic lymph node dissection, with the same criteria for the execution of lymphadenectomy as reported by the previous study. No statistically significant differences were found between the two groups in terms of OS and DFS. The authors concluded that systematic mediastinal lymphadenectomy is not necessary for the treatment of clinical peripheral NSCLC < 2 cm in diameter.

Subsequently, Wu et al. [34] conducted a prospective randomized study on the comparison of systematic lymph node dissection and lymph node sampling in the treatment of pathological stage I-IIIA NSCLC patients. In the lymph node sampling group, nodes of regions 1–9 were explored and any nodes with suspected cancer metastases were removed, while nodes of station 7 were excised routinely in all patients. Trial results showed more stage I and fewer stage IIIA in the lymph node sampling group, suggesting that this technique is less accurate than complete dissection in the cancer staging. The systematic lymphadenectomy group had a statistically significantly better OS and local control with a higher DFS than the sampling group, allowing the eradication of the locally advanced disease with an improvement in long-term survival. Therefore, systematic mediastinal lymphadenectomy with lobectomy is recommended for the treatment of stage I-IIIA NSCLC, being a safe procedure that improves survival.

The more recent ACOSOG Z0030 Trial [35] was a prospective, randomized, controlled, multi-institutional study, aimed to evaluate the outcomes of mediastinal lymph node dissection compared to mediastinal lymph node sampling in the treatment of N0 or non-hilar N1, T1-T2 NSCLC patients, in terms of survival and recurrence patterns. Patients diagnosed with proven NSCLC, before randomization, underwent rigorous mediastinal and hilar lymph node sampling: lymph node stations 2R, 4R, 7, and 10R were sampled for tumors in the right lung; instead, stations 5, 6, 7, and 10L were sampled for tumors in the left lung. Any other suspicious lymph nodes were also biopsied. If all lymph nodes sampled were negative at frozen section examination, patients were intraoperatively randomized into the sampling group, in which no extra lymph nodes were removed, or into the complete systematic lymphadenectomy group. In line with the results of this trial, the systematic lymphadenectomy does not improve survival or affect local or regional recurrence rate in early-stage NSCLC, with negative pathologic hilar and mediastinal lymph nodes after surgical staging. However, the available preoperative diagnostic methods are not able to exclude the presence of lymph node metastases like surgical staging, and therefore the authors still recommend performing systematic mediastinal lymphadenectomy, since it does not increase morbidity and mortality.

To investigate the efficacy of lobe-specific lymphadenectomy compared with complete lymphadenectomy, Okada et al. [36] performed a prospective non-randomized study on the surgical treatment of clinical stage I NSCLC. Selective dissection of upper mediastinal nodes, including aortic regions, was planned in case of upper lobe tumor with hilar and lower mediastinal nodes negative; the dissection of the lower mediastinum in patients with a lower lobe tumor with hilar and upper mediastinum nodes negative was performed. This study demonstrated the non-inferiority of lobe-specific lymphadenectomy in terms of OS and DFS at 5 years, compared to complete lymphadenectomy, which was related to increased morbidity.

Another prospective non-randomized study investigating the role of selective lymph node dissection has been conducted [37], comparing the results of patients treated with lobectomy and systematic lymphadenectomy or lobe-specific lymph node dissection, for clinical and intraoperative (based on the surgeon’s impressions or/and intraoperative histological examination) N0 NSCLC. Patients undergoing selective lymphadenectomy were divided into two groups: patient-related risk factors, including advanced age and severe diabetes, respiratory dysfunction or cardiovascular disease, and tumor-related factors, including C/T ratio of <0.5, SUVmax < 2.5 on FDG-PET and elevation of serum tumor markers. In the group of patients who underwent lobe-specific lymphadenectomy due to tumor-related factors, the 5-years DFS and OS were 100%, so the authors suggested that the lymph node sampling may have been proper in these patients. The group who underwent lobe-specific lymphadenectomy because of patient-related risk factors showed no significant differences in the 5-DFS and 5-OS compared with systematic lymphadenectomy, but significantly higher initial recurrence of mediastinal node cancer.

To date, only one prospective randomized clinical trial has been concluded on the comparison between systematic and lobe-specific lymphadenectomy in patients with stage T1aN0M0 (<2 cm) NSCLC [38]. This study included 96 patients who underwent radical lung resection and then randomized into the two methods of lymph node evaluation. The results showed that lobe-specific lymphadenectomy resulted in a shorter length of stay, blood loss, and postoperative complications. No statistically significant differences were found between the two groups in terms of survival, recurrence, and N-state migration. Furthermore, it emerged that in cases of a lesion with a high rate of GGO, no lymph node metastasis occurred, therefore in these patients it may not be necessary to perform the systematic assessment.

In January 2017, a randomized Phase III trial was designed by the Japan Clinical Oncology Group-Lung Cancer Surgical Study Group (JCOG-LCSSG) to evaluate the clinical benefit in terms of survival non-inferiority and less invasiveness of lobe-specific lymphadenectomy, compared with systematic nodal dissection, in patients with clinical Stage I–II NSCLC [39], and we are looking forward the result to clarify this issue.

**Table 2 cancers-15-03735-t002:** Selected randomized and prospective non-randomized studies investigating the role of lymphadenectomy in early-stage NSCLC.

Investigators	Year	N of Patients	Stage	Results
Izbichi et al. [28]	1998	169	I-IIIA	Systematic lymphadenectomy does not improve OS and DFS, compared to sampling in pN0 patients, it seems to slightly improve DFS in pN1 and pN2
Sugi et al. [33]	1998	115	cT1a-bN0M0	No difference in terms of survival and recurrence between systematic and sampling, demonstrating that peripheral tumors < 2 cm do not require hilum-mediastinal lymphadenectomy
Wu et al. [34]	2002	532	cI-IIIA	Systematic lymph node dissection improves survival and DFS, compared with sampling in clinical stage I-IIIA NSCLC
Okada et al. [36]	2006	735	cI	Lobe-specific lymphadenectomy is non-inferior to systematic in clinical stage I NSCLC in terms of DFS and OS
Darling et al. [35]	2011	1023	pT1-T2N0-N1M0	If systematic hilar and mediastinal sampling is negative, systematic lymphadenectomy does not improve survival in early-stage NSCLC
Maniwa et al. [37]	2013	335	N0	The recurrence of mediastinal node cancer in patients undergoing lobe-specific lymphadenectomy was significantly greater than that in those undergoing systematic dissection
Ma et al. [38]	2013	96	cT1aN0M0	Lobe-specific lymph node dissection is similar to systematic in terms of migration of N stage, OS, and DFS, with fewer postoperative complications, bleeding, and length of stay
Hishida et al. [39]	2017	1700	cI-II	Ongoing trial

### 2.3. Review and Meta-Analysis

The principal reviews and meta-analyses conducted, over the years, on the role of hilar-mediastinal lymphadenectomy in the treatment of early-stage NSCLC have provided controversial and sometimes inconclusive results (Table 3).

On the comparison between systematic mediastinal lymph node dissection and mediastinal lymph node sampling in the treatment of pathological stage I NSCLC, Dong et al. [40] performed a meta-analysis of the scientific evidence reported in the literature until 2014. The authors considered comparative studies evaluating the survival rate at 1, 3, and 5 years, demonstrating that there is no statistically significant difference in the 1-year survival between the two techniques, while at 3 and 5 years the systematic lymphadenectomy is superior in terms of survival rate.

In the same year, Huang et al. [41] published a systematic review and meta-analysis about the role of systematic dissection and sampling in the management of stage I-IIIA NSCLC, considering randomized controlled trials. In terms of OS and local and distant DFS, systematic mediastinal lymphadenectomy and mediastinal lymph node sampling appeared similar in early-stage NSCLC patients, whereas it was not clear if complete lymphadenectomy is superior for stage II-IIIA.

In 2017, Mokhles et al. [42] conducted a systematic review and meta-analysis of randomized controlled trials on the comparison of systemic mediastinal lymph node dissection and mediastinal lymph node sampling during lobectomy for NSCLC, to review lymph node dissection recommendations. The conclusions from the analysis of these trials are uncertain given the high risk of bias, suggesting that large pragmatic multicenter studies need to be conducted to obtain reliable recommendations.

Since the benefits of different types of lymph node dissection in the treatment of early-stage NSCLC were unclear, a further systematic review and meta-analysis was published, in 2016 [43], on the comparison of survival between systematic lymph node dissection, lymph node sampling, and lobe-specific lymph node dissection in the treatment of clinical early-stage NSCLC. Lobe-specific lymphadenectomy seemed associated with the same results in terms of survival as the systematic one in the treatment of the early stages, while sampling appears inferior to the other methods.

In conclusion, in 2021, Luo et al. [44] published the results of a meta-analysis on the role of selective and complete lymphadenectomy in the management of clinical stage I NSCLC. This meta-analysis found that lobe-specific lymphadenectomy is equivalent to systematic mediastinal lymphadenectomy in terms of survival and disease control, but is associated with fewer postoperative complications. They conclude that selective lymphadenectomy should be the treatment of choice for clinical stage I NSCLC.

**Table 3 cancers-15-03735-t003:** Selected review and meta-analysis investigating the role of lymphadenectomy in early-stage NSCLC.

Investigators	Year	N of Patients	Results
Dong et al. [40]	2014	711	In pathological stage I NSCLC, sampling vs. systematic lymphadenectomy are equal in 1-year survival rate, better for systematic at 3 and 5 years
Huang et al. [41]	2014	1791	In terms of OS and DFS, systematic lymph node dissection does not differ from sampling in stage I-IIIA NSCLC
Meng et al. [43]	2016	3955	Lymph node sampling is inferior in terms of survival for early-stage NSCLC, lobe-specific and systematic are equal
Mokhles et al. [42]	2017	1980	The high risk of bias in these trials makes the overall conclusion insecure
Luo et al. [44]	2021	5713	Selective mediastinal dissection is preferable in stage I NSCLC, with the same survival and control of local and distant disease and fewer postoperative complications

## 3. Discussion

Although it is widely acknowledged that lymph node assessment is essential for accurate staging in the treatment of resectable NSCLC, the extent of lymph node dissection and its impact on survival are controversial. The international guidelines give conflicting indications on the number of mediastinal lymph nodes necessary to remove and on the most suitable type of lymphadenectomy to perform concerning the stage [4,5,6,7,8,9].

Systematic lymphadenectomy has been considered the gold standard treatment since 1951, when Cahan reported: “that some patients experienced long-term survival when the positive regional lymph nodes also were removed” [45]. However, the standard treatment is not respected by many surgeons in clinical practice. In a review on the surgical treatment of NSCLC in the United States, Little found that only 57.8% of patients undergoing surgery have any mediastinal lymph node sampled or removed, and in community hospitals this percentage decreases even more (48.1%) [46]. Hilar-mediastinal lymphadenectomy may result in a technically challenging procedure, particularly in the case of less experienced thoracic surgeons and video-assisted lung surgery [36], resulting also in prolonged and stressful operations. However, in contrast with what happens in clinical practice, complete lymphadenectomy remains recommended in all cases of resectable NSCLC [13], even though there are also two other popular methods to assess the lymph node status: mediastinal lymph node sampling and selective mediastinal lymphadenectomy.

With the increasing detection of clinical stage I NSCLC, thanks to the screening programs and the improvement of diagnostic-staging pathways [10,11,12], the key question is whether systematic mediastinal lymphadenectomy is always necessary. Arguments in favor are the more accurate staging with access to adjuvant therapies and better control of loco-regional recurrences, against the risk of greater morbidity and mortality, related to a more complex procedure in a supposed local stage of the disease [40].

To date, with this narrative review, it does not seem possible to conclude the appropriate type of lymph node evaluation in this subgroup of patients.

### 3.1. Limits of Previous Studies

The most relevant studies on the role of lymphadenectomy in the early-stage NSCLC exposed in this narrative review have several critical limitations, which could account for such conflicting results on this topic. The previously published studies, even the most recent ones, include indeed heterogeneous samples, in terms of stages and lymphadenectomy technique, that are difficult to compare, making the evaluation of the results and the conclusions complex.

First, the surgical methodology of lymphadenectomy, in its existing variants, is not standardized [16], therefore the comparison between the different studies and their results is not effectively achievable. For instance, the type of lymphadenectomy to be performed is not based on objective and reproducible criteria, but it depends mainly on the surgeon’s preference and the patient’s condition [15]. The lack of detail in relation to the partial or total removal of the lymph nodes [15] and the number of lymph nodes removed at the different stations [21] is another criticism of several reported studies. Furthermore, in the different articles, the authors used the appellation “systematic” to define lymphadenectomy with different extent: from the resection of at least two mediastinal stations always including station 7 [25] to the removal of at least 10 lymph nodes with exploration of at least two mediastinal stations [16]. For these reasons, many systematic reviews and meta-analyses conclude indeed that there are too many discrepancies among centers on lymphadenectomy procedures and policy [40,41,42,43].

In addition, to not being standardized, the lymphadenectomy techniques proposed in some studies have created several criticisms, as happened in the ACOSOG Z0030 randomized trial [35]. Systematic surgical sampling of the mediastinum performed in this study, before randomization with frozen section on the lymph nodes removed, is a very effective and quality method, resulting in reliable results, but is rarely achieved to the detriment of patient outcomes [47]. Furthermore, these results do not improve clinical practice since surgical staging is not comparable to clinical staging. If on the one hand this method guarantees to adequately select patients in the early stage, on the other hand it excludes from the randomization those patients who are clinically N0, but pathologically N+, nullifying the principle of intention to treat [42].

Many authors instead have included sublobar anatomical resections and atypical resections in the sample, which have different survival rates compared to lobectomy, producing bias in the results [15,35,43].

Another factor that emerges from the analysis of the literature is that many articles do not include only the early stages, i.e., stage I, but also locally advanced ones [32,34,35,39], involving various confounding factors due to the significant heterogeneity of the sample, with their related different prognosis. Even the most recent randomized study on the comparison between systematic and lobe-specific lymphadenectomy currently ongoing also includes stage II NSCLC and not just stage I [39].

Precise preoperative clinical staging of patients with non-small cell lung cancer is essential to define the appropriate treatment strategy. The preoperative lymph nodal evaluation was based on CT scan in several studies published before the introduction of PET in clinical practice [15,16,18,20,28,33,34,38], with a consequent reduction in the precision of clinical staging and increased risk of enrolling patients in more advanced stages. Improvements in CT scan technology have allowed the morphological definition of the lesions in an ever more accurate way, but this investigation has limitations in defining the nature and the progress of the lesion found. The PET-CT 18-FDG, combining the morphological information of the CT with the functional criterion of the PET, has made the staging of lung cancer more accurate, allowing it also to evaluate the lymph node status [48,49]. PET-CT was introduced into clinical practice in 1998 [50] and it was not used in preoperative staging in many of the aforementioned studies, resulting in inadequate patient staging and selection [42].

Finally, two systematic reviews and meta-analyses criticize that different papers do not mention the use of postoperative adjuvant chemotherapy [42,43], which is currently the standard of care for stage III, which for understandable reasons impacts oncological outcomes [4,5,6].

### 3.2. Future Perspective

It is necessary to conduct future prospective randomized trials that meet certain criteria to define scientific recommendations, which could guide clinical practice in the management of early-stage lymphadenectomy:-Inclusion of unique clinical stage I NSCLC;-Adequate preoperative staging, based on the most recent guidelines;-Define lymphadenectomy criteria that can be reproduced easily in subsequent studies, such as the number of lymph nodes removed and the corresponding lymph node stations, rather than the type of lymphadenectomy;-Homogenize the sample by the type of lung resection.

While surgical treatment in the early stage is considered the treatment of choice, 30% of patients manifest recurrence, mainly distant ones [16]. A possible explanation could be the presence of occult micrometastases in the regional mediastinal lymph nodes, which in 20.4-44.9% of the cases are found in pathological N0 stage I NSCLC patients by immunohistochemistry or molecular methods [51,52], with negative histopathological examination.

It has long been widely recognized that N status is the most significant prognostic factor impacting survival in NSCLC [53] and, to date, it has become necessary to define the factors associated with lymph node metastases and how to detect early occult lymph node micrometastases. Koike sought to identify the population at high risk of lymph node metastasis among patients with clinical stage IA [54] and defined predictor factors such as age ≤ 67, preoperative serum carcinoembryonic antigen level ≥ 3.5 ng/mL, tumor size on preoperative radiologic findings ≥ 2 cm, and consolidation/tumor ratio on high-resolution computed tomography ≥ 89. Future studies are expected, which could allow the identification of high-risk early-stage patients for lymph node metastases in the perioperative period [16], who could benefit from a more aggressive lymph node dissection.

Occult lymph node micrometastases are defined as lymph nodes with metastatic cells that previously resulted negative at the preoperative staging and conventional histopathological methods; their presence is considered a marker for primary tumors with high metastatic potential [54,55,56]. As reported by Bille et al. [25], the postoperative finding of occult N2 micrometastases is associated with a lower survival rate and a higher risk of metastatic disease compared to N0 patients and is the main prognostic factor [57].

Improving the detection of occult lymph node micrometastases, especially N2, could have an important prognostic role in lung cancer staging, guiding the treatment pathway. To date, several molecular biology methods are available for the detection of micrometastases in NSCLC [58]: the polymerase chain reaction (PCR), immunohistochemistry (IHC), flow cytometry (FCM), immunomagnetic beads (IMB), and one-step nucleic acid amplification (OSNA). 

PCR is widely used in molecular biology [59] and the most widespread in the detection of micrometastases is reverse transcription (RT-PCR). Recently, digital PCR (dPCR) is being used in the detection of lymph node metastases from breast and colorectal cancer [60,61]. The most common markers are human lung-specific X (LUNGX) protein [59], epidermal growth factor receptor (EGFR) protein, and human telomerase reverse transcriptase (hTERT) [62]. A new biomarker and therapeutic target for lung cancer metastases, discovered by Lian et al. [63], is DDX49 in the Akt/B-catenin pathway.

IHC is a widespread method for detecting lymph node micrometastases based on antigen-antibody responders and a low CDH1/CDH2 ratio has recently been shown to be associated with lung cancer micrometastases [64].

The OSNA method detects CK19 RNA copy number and has been shown to be more sensitive than hematoxylin-eosin staining in detecting lymph node micrometastases in NSCLC [65]. 

Several studies recommend carrying out the frozen-section evaluation of the lymph nodes [35] or T intraoperatively [20], to define which type of lymphadenectomy to perform, but this evaluation is currently highly operator-specific, greatly increasing the intraoperative times. The introduction of a methodology such as OSNA, that can provide reliable non-operator-related results in a short time, could be a turning point in deciding intraoperatively the extent of lymphadenectomy.

Finally, the detection of circulating tumor cell DNA (ctDNA) for the presence of early NSCLC in liquid biopsy has been shown to be effective and associated with several biomarkers, but there are still gaps in the study of related biomarkers for lymph node micrometastases. 

It is necessary to direct research towards the detection of highly specific and sensitive tumor markers of micrometastases, to achieve a more accurate pathological staging and to define future perspectives.

The major limit for precise pathological nodal staging is represented by the surgeon who decides not to perform lymphadenectomy [35,42]. Probably, this will be overcome also thanks to new technologies with the integration of imaging, allowing to perform the procedure in safety and comfort, working in an ergonomic way (for example with the integration of imaging in robotic systems).

## 4. Conclusions

If we want to reduce the incidence of recurrences, without performing unnecessary at-risk procedures in the early stages, we need to establish the strategies for an adequate lymphadenectomy, to give an easier option for the surgeon that guarantees positive results in a subset of patients.

Today, there is no standardization on reproducible lymphadenectomy and most of the studies conducted do not include only stage I NSCLC.

It is advisable to proceed with studies on the intrinsic characteristics of the tumor and on the mechanism of its dissemination, to investigate the risk factors related to lymph node metastatic involvement and clarify this issue.

Prospective randomized studies including only clinical stage I NSCLC are required to establish the outcomes stratified by the characteristics of the tumor. Furthermore, an evaluation of the risk factors related to possible worse oncological outcomes of incomplete lymphadenectomy is necessary, to consequently make recommendations on the lymph node assessment in early-stage patients.

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
