# Peer review of "The Role of Lymphadenectomy in Early-Stage NSCLC"

_cancers, 2023, doi:10.3390/cancers15143735_

Round 1

Reviewer 1 Report

  • The manuscript clear, relevant for the debated fiel of intraoperaive lyphadenectomy in lung cancer surgery. The manuscript is well structured, highlighting the importance of the topic. The cited references are relevant, only some are not from the last 5 years, but their presence is well structured mostly in the historical presentation of the problems of this field. 
  • The manuscript has no self citations.
  • The manuscript logically summarizes some recent studies to clarify the importance of lymphadenectomy in NSCLC patients, and the controversies of this field. 
  • The tables can clearly demonstrate the results of studies dealing with the vary problem, there are no specific images available. 
  • Data are clearly presented and interpreted consistently during the whole manuscript. It is easy to understand ad they clarly demonstrate the findings cited in the manuscript. 
  • The conclusions are consistent with the evidence and arguments presented

Qulity of english is overall good, it is easy to understand and clear to read.

Author Response

Answer for Reviewer 1

Comments and Suggestions for Authors

The manuscript clear, relevant for the debated fiel of intraoperaive lyphadenectomy in lung cancer surgery. The manuscript is well structured, highlighting the importance of the topic. The cited references are relevant, only some are not from the last 5 years, but their presence is well structured mostly in the historical presentation of the problems of this field. 

The manuscript has no self citations.

The manuscript logically summarizes some recent studies to clarify the importance of lymphadenectomy in NSCLC patients, and the controversies of this field. 

The tables can clearly demonstrate the results of studies dealing with the vary problem, there are no specific images available. 

Data are clearly presented and interpreted consistently during the whole manuscript. It is easy to understand ad they clarly demonstrate the findings cited in the manuscript. 

The conclusions are consistent with the evidence and arguments presented.

Thank you very much for the appreciation of our paper.

Reviewer 2 Report

Dear authors, 

this is a well written narrative review of lymphadenectomy in (early stage) lung cancer.

I have 3 comments:

While all the cited studies are well known for years it would be interesting to go a little bit more into details concerning the limitations of each study and provide a more critical discussion. 

Especially the question, if systematic lymph node dissection is associated with more complications than a thorough and systematic sampling remains unclear. 

The difference between systematic dissection of lymph node stations and sampling of nodes is the remaining network of lymphvessels in the mediastinal fatty tissue. Did you find any references in the literature on this topic?

Finally, in the long list of references, I miss the paper prom Liang on the importance of the number of lymph nodes dissected, which refers to a more radical systematic dissection. (J Clin Oncol 35:1162-1170)

I congratulate to this fine work and I am thankful for the privilege to review this paper. I hope, the comments are helpful. 

Some formulations are cumbersome, but otherwise the language is ok

Author Response

Answer for Reviewer 2

Comments and Suggestions for Authors

Dear authors, 

this is a well written narrative review of lymphadenectomy in (early stage) lung cancer.

Thank you very much for the appreciation of our paper and for the useful, interesting, and conscientious suggestions. I will reply to comments in sequence.

I have 3 comments:

  1. While all the cited studies are well known for years it would be interesting to go a little bit more into details concerning the limitations of each study and provide a more critical discussion. 

Thanks very much for the suggestion. We consciously chose not to go into detail on the limitations of each study because, in our opinion, the review could become long-winded.

  1. Especially the question, if systematic lymph node dissection is associated with more complications than a thorough and systematic sampling remains unclear. 

Thank you very much for the helpful comment. I agree that the association between systematic lymphadenectomy and major complications is a controversial issue to date. For this reason, we have chosen not to delve into this issue in order to avoid the risk to lose the focus of the study (the survival and recurrence associated with lymphadenectomy in the early-stage NSCLC), already a very structured and wide topic.

  1. The difference between systematic dissection of lymph node stations and sampling of nodes is the remaining network of lymphvessels in the mediastinal fatty tissue. Did you find any references in the literature on this topic?

Thanks for this interesting observation. We have found no reference to this issue in the literature. It will certainly be an exciting topic to explore in the future.

  1. Finally, in the long list of references, I miss the paper prom Liang on the importance of the number of lymph nodes dissected, which refers to a more radical systematic dissection. (J Clin Oncol 35:1162-1170)

Thank you very much for your comment. When evaluating retrospective studies in the literature, we chose to focus on studies that considered early-stage NSCLC to homogenize the analysis as much as possible. We therefore decided to omit the paper you mentioned because it considered stages I-IIIA, which could have been confusing.

I congratulate to this fine work and I am thankful for the privilege to review this paper. I hope, the comments are helpful. 
